# Porous TiO_2_/Carbon Dot Nanoflowers with Enhanced Surface Areas for Improving Photocatalytic Activity

**DOI:** 10.3390/nano12152536

**Published:** 2022-07-23

**Authors:** Fengyan Song, Hao Sun, Hailong Ma, Hui Gao

**Affiliations:** 1Center of Excellence for Environmental Safety and Biological Effects, Beijing Key Laboratory for Green Catalysis and Separation, Department of Chemistry and Biology, School of Life Science and Chemistry, Faculty of Environment and Life, Beijing University of Technology, Beijing 100124, China; fengyansong@bjut.edu.cn; 2Ningbo Institute of Technology, Beihang University, Ningbo 315100, China; hao.sun@buaa.edu.cn (H.S.); hailongma@buaa.edu.cn (H.M.); 3School of Aeronautic Science and Engineering, Beihang University, Beijing 100191, China; 4Hangzhou Innovation Institute (Yuhang), Beihang University, Hangzhou 310052, China

**Keywords:** porous TiO_2_, hybrid TiO_2_/CDs, photocatalysts, photodegradation, large surface areas

## Abstract

Electron–hole recombination and the narrow-range utilization of sunlight limit the photocatalytic efficiency of titanium oxide (TiO_2_). We synthesized carbon dots (CDs) and modified TiO_2_ nanoparticles (NPs) with a flower-like mesoporous structure, i.e., porous TiO_2_/CDs nanoflowers. Among such hybrid particles, the CDs worked as photosensitizers for the mesoporous TiO_2_ and enabled the resultant TiO_2_/CDs nanoflowers with a wide-range light absorption. Rhodamine B (Rh-B) was employed as a model organic pollutant to investigate the photocatalytic activity of the TiO_2_/CDs nanoflowers. The results demonstrated that the decoration of the CDs on both the TiO_2_ nanoflowers and the (commercially available AEROXIDE TiO_2_) P25 NPs enabled a significant improvement in the photocatalytic degradation efficiency compared with the pristine TiO_2_. The TiO_2_/CDs nanoflowers, with their porous structure and larger surface areas compared to P25, showed a higher efficiency to prevent local aggregation of carbon materials. All of the results revealed that the introduced CDs, with the unique mesoporous structure, large surface areas and loads of pore channels of the prepared TiO_2_ NPs, played important roles in the enhancement of the photocatalytic efficiency of the TiO_2_/CDs hybrid nanoflowers.

## 1. Introduction

Pollutants such as inorganic nitrogen oxides and organic dyes from gas or liquid phases may cause severe environmental and health problems; hence, complete degradation and elimination of them is of great importance [1,2,3]. The removal of pollutants via a photocatalytic oxidation process using semiconductor materials and harnessing solar energy has been gaining increasing interest in the last decade because it is a promising green technology [4,5]. Titanium oxide (TiO_2_) is anticipated as a promising material for photocatalytic reactions due to its low toxicity, high chemical stability, availability, abundance and low cost [6,7,8,9]. However, the application of pure TiO_2_ is limited by its relatively low solar photo-conversion efficiencies, because its wide band gap (3.2 eV for anatase and 3.0 eV for rutile) requires photocatalytic activation only by ultraviolet irradiation (<4% of the total solar spectrum) [10], and the high electron–hole recombination rate in the TiO_2_ particles results in low efficiency of photocatalytic reactions [11]. Therefore, great efforts have been directed towards modifying the TiO_2_ to extend its photocatalytic activity into the visible region and suppress the recombination of photo-generated electron–hole pairs, thereby enhancing solar energy conversion and improving the photocatalytic efficiency. To achieve this target, various strategies have been adopted, including doping with transition metal ions (e.g., V, Cr, Fe) [12], coupling with narrow band gap semiconductors [13,14], depositing noble metals [15,16], and incorporating non-metal elements [3,6,17,18,19]. Doping TiO_2_ with metal ions narrowed the band gap due to the formation of intermediate energy levels in the band gap [6]. However, the metal-doped TiO_2_ normally suffers from thermal instability and atom diffusion, which results in a low photocatalytic efficiency [12]. Non-metal doping of TiO_2_ has since been proved to be a robust material, and many reports on TiO_2_ doped with B, F, N, S, I and C have proved a significant improvement in the visible-light photocatalytic efficiency [1,11,20,21].

The decoration of TiO_2_ with carbonaceous nanomaterials such as carbon nanotube [22], fullerene (C60) [23,24], graphene [25,26], etc., is being increasingly investigated. The resultant TiO_2_/carbon composites were demonstrated in the literature with an improved optical absorbance and better photocatalytic activities in the visible light range when compared with pristine TiO_2_ [6,27]. Carbon dots (CDs) and 0-D (zero-dimensional) nanodots have recently emerged as new members of the nano-carbon materials, and have drawn great attention due to their unique properties [28,29]. Like traditional semiconductor quantum dots, CDs show excellent photoluminescence, great resistance to photo-bleaching and good chemical stability. In addition, they possess the advantages of well-water dispersibility, non-toxicity, electron-accepting and transport properties, and facile production of a low cost [30,31]. To date, CDs have been applied to couple with TiO_2_ to extend their light absorption to a visible range for better photocatalytic performance. Researchers believed that CDs can improve optical absorption, trap charge carriers, and hinder the recombination of the photo-generated electron–hole pairs [32,33,34]. In addition, CDs with up-conversion properties may even broaden their absorption to a (near-infrared) NIR light range and enable them with NIR photocatalytic activity [35]. However, up until now, most of the reported TiO_2_/CDs photocatalysts were mainly focused on employing TiO_2_ particles with nanotubes, nano-belts, or spherical morphology [32,35,36,37]. These TiO_2_ nanocomposites normally possess low surface areas, which tends to cause uneven distribution and local aggregation of carbon materials, making them the recombination center of photo-generated carriers and thus reducing the utilization of light [11].

To solve this problem, we introduced wide visible-to-NIR absorption range CDs into the mesoporous TiO_2_ matrix, with a precise pore size of around 5 nm to simultaneously utilize the advantages of the CDs and the mesoporous-structured materials with high surface areas. The porous structure and high surface areas may increase the effective contact areas between the pollutants and the active surfaces of photocatalytic particles, thus reducing the local CDs aggregate, which can significantly improve catalytic efficiency [22]. Herein, TiO_2_, with its mesoporous structure and higher surface areas than the (commercially available AEROXIDE TiO_2_) P25, was synthesized using the hydrothermal and calcination methods. CDs were fabricated by the simple hydrothermal carbonation of biomass and were then introduced into the mesoporous TiO_2_ particles by a self-assembly process, thereby forming TiO_2_/CD hybrid particles. The photocatalytic behavior of the as-synthesized TiO_2_ and TiO_2_/CD samples towards the degradation of rhodamine B (Rh-B) were studied and exhibited higher catalytic efficiency compared to the commercial P25 TiO_2_.

## 2. Materials and Methods

### 2.1. Preparation of TiO_2_ Mesoporous Particles

A total amount of 2 mL of titanium butoxide was dispersed in a solution of 20 mL of NH_3_H_2_O (37%) dissolved with 0.28 g KCl. The suspension was stirred for 30 min, transferred to a Teflon-sealed autoclave reactor (Yan Zheng Instrument, Shanghai, China) and maintained at 180 °C for 4 h. The precipitates were then washed with ethanol and DI (deionization) water a few times to remove free ions, and then freeze-dried. Finally, the resulting dry powder was calcined by a conventional method, in which the calcinations was conducted at a temperature of 550 °C for 4 h in air with a heating ramp of 10 °C/min.

### 2.2. Preparation of Carbon Dots

A total amount of 0.7 g D-(+)-glucose was dissolved in 20 mL ethanol and placed in a Teflon-lined, stainless-steel autoclave, which underwent treatment at 200 °C for 6 h. The dark brown and yellow solution obtained was centrifuged at 10,000 rpm for 20 min to remove the solution containing fluorescent CDs from the solid black precipitate. The solution of CDs was then filtered using standard syringe filters.

### 2.3. Preparation of TiO_2_/CDs Photocatalysts

The as-prepared mesoporous TiO_2_ particles were coated with the CDs by soaking them in the CD solution in ethanol for 24 h, before removing and rinsing with clean ethanol to remove any poorly adhered CDs.

### 2.4. Characterization of the Photocatalysts

Several techniques were employed for the characterization of the samples. In order to determine the crystal phase composition of the as-prepared TiO_2_ particles, X-ray diffraction (XRD) measurements were carried out at room temperature using a Bruker D5000 (Bruker, Karlsruhe, Germany) with Cu Kα radiation in the region of 2θ = 15–70°. The morphology and surface characteristics of the TiO_2_ particles were investigated using Scanning Electron Microscopy (SEM, FEI Inspect-F, Hillsboro, OR, USA). The samples were observed using an accelerating voltage of 20 kV, a spot size of 3.5, and a working distance of approximately 10 mm. The microcrystalline morphology and structure of the samples was analyzed by Transmission Electron Microscopy (TEM) and High-Resolution (HR) TEM, using a JEOL JEM-2010 electron microscope (Tokyo, Japan) operating at 200 kV. The BET surface area determination was obtained by measuring the N_2_ adsorption–desorption isotherm using an Autosorb-IQ2-MP-C system (Quantachrome Instruments, Boynton Beach, FL, USA). During the Brunauer–Emmett–Teller (BET) analysis, the samples were degassed at 150 °C for 24 h prior to the nitrogen adsorption measurements. The specific surface area was calculated by the multipoint BET method, and the pore-size distribution was calculated based on the Quenched Solid Density Function Theory (QSDFT) using the adsorption branch. Absorption spectra of the products (CDs, TiO_2_, and TiO_2_/CDs hybrid particles) were measured using a PerkinElmer Lambda LS 35 UV/Vis spectrometer (Waltham, MA, USA) The fluorescence spectra of the CDs and the TiO_2_/CDs were measured using the fluorescence spectrometer PerkinElmer LS 55, with a slit width of 10 nm both for excitation and emission. Fourier transform infrared (FTIR) spectra were recorded on an FTIR spectrometer 100 (PerkinElmer), collecting data canning from 4000 to 400 cm^−1^ at a spectral resolution of 4 cm^−1^.

### 2.5. Photocatalytic Degradation Experiments

The photocatalytic activity of the synthesized pure TiO_2_ and TiO_2_/CDs was tested under a Xenon lamp (λ > 340 nm, ~110 mW cm^−2^). A Rh-B solution with a concentration of 20 µg/mL was prepared in DI water. Typically, 10 mg of powder samples were suspended in 30 mL of a Rh-B solution. Prior to the light irradiation, the suspension was kept in the dark for 30 min under magnetic stirring to reach the adsorption–desorption equilibrium between the Rh-B molecules and the photocatalysts. Then, the above suspension was kept in a quartz cuvette, which was exposed to the irradiation light (λ > 340 nm). An ice bath was applied to ensure that the temperature change of the suspensions was less than 5 °C. With magnetic stirring, 3 mL of the dispersion was taken out at regular intervals and centrifuged at 10,000 rpm for 5 min, and then the supernatant solution was collected and analyzed using a UV-Vis (ultraviolet-visible) absorbance spectroscopy. For comparison, the photocatalytic reactions were carried out with the catalyst of the P25 and the P25/CDs under the same procedure. The percentage of degradation was calculated as C_t_/C_0_, where C_t_ is the concentration of the remaining dye solution at each irradiated time interval, while C_0_ means the concentration of the Rh-B solution after keeping it in the dark for 30 min in the presence of any photocatalyst.

## 3. Results and Discussion

As depicted in Figure 1, the synthesis of the porous TiO_2_/CDs photocatalyst was realized with a few steps. The TiO_2_ with a porous structure was fabricated through a combined hydrothermal reaction (180 °C) and a calcination method (550 °C) using titanium butoxide, KCl and NH_4_OH as starting materials. The TiO_2_ allows a controlled interface and nanocrystal growth under such high thermal treatment. The CDs were produced by the hydrothermal carbonization of D-(+)-glucose at 200 °C and were simply introduced into porous TiO_2_ by a self-assembly method. The formed TiO_2_/CDs were then used for a degradation study of organic dyes.

The XRD measurement of the as-prepared TiO_2_ particles was carried out to investigate their crystalline structure, as shown in Figure 1a. The XRD pattern exhibited strong diffraction peaks at 25.4°, 37.8°, 48°, 53.9°, 55.1° and 62.7°, which correspond to (101), (004), (200), (105), (211), and (204) faces of anatase TiO_2_ [38]. Notably, there were no apparent peaks at the positions of 27.58°, 41.38, and 44.18, which are the characteristic peaks for rutile TiO_2_. It shall be mentioned that anatase and rutile are two types of TiO_2_ polymorphs that are typically used for photocatalytic application. The anatase structure is preferred due to its significantly higher photocatalytic activity than the rutile phase for several reasons: (1) anatase has a larger band-gap than rutile, which provides anatase with a higher redox potential; (2) anatase has a higher area density of surface hydroxyls. This high area density of surface hydroxy favors capturing the CDs and slows the recombination of photogenerated electron–hole pairs [22,28]. Even though the real causes of the better photocatalytic properties of anatase are not yet fully understood, it is speculated that the charge-carrier mobility in anatase is 89 times faster than that in rutile, which is around 80 cm^2^ V^−1^ s^−1^ [22,39,40]. Hence, we expected that the as-synthesized TiO_2_ particles that were mainly composed of anatase phase TiO_2_ may possess a superior photocatalytic activity. The SEM image shows porous and aggregated particles, which were self-assembled by petal-shaped TiO_2_ nanoparticles (Figure 2a), while the TEM image in Figure 1b displays a network of porous interconnected channels due to these constituent-agglomerated petal-shaped TiO_2_ nanoparticles, demonstrating a sponge-like and mesoporous architecture. The HR-TEM image of the TiO_2_ materials shows clear lattice fringes of the TiO_2_ particles, indicating an interplanar spacing of 0.35 nm (Figure 1c), which matched well with the (101) plane of the anatase TiO_2_. In addition, our HRTEM image indicates that the crystallite of the TiO_2_ is indeed nano-sized, and the aggregated particles were with the pore structures formed by these nanocrystals, as can be verified from the contrasts in the HRTEM image (Figure 1c). The crystallographic structure of the TiO_2_ material was further characterized by selected area electron diffraction (SAED), as displayed in Figure 1d. With a bright 101 ring, the diffraction rings can be indexed perfectly to the anatase phase of the TiO_2_ [41]. Both the HRTEM and SAED analyses demonstrated perfectly crystallized TiO_2_ materials, which corresponded well with the XRD results.

When CDs were introduced into the porous TiO_2_ by a self-assembly method, the porous TiO_2_/CDs nanoflower structures formed (Figure 2b). The nanoflower structures can enlarge porosity of the TiO_2_/CD hybrid particles. Such TiO_2_/CD hybrid particles with large porosity normally possess high surface areas, which tends to increase the effective contact areas between the pollutants and the active surfaces of the photocatalytic particles. To confirm that the porous TiO_2_ are incorporated with the CDs, Energy Dispersive X-ray (EDX) spectrometry was conducted. Peaks of C, O and Ti elements are shown in the EDX spectra of the hybrid TiO_2_/CD nanoflowers (Figure 3), in which C is from the assembled CDs, demonstrating that the CDs were successfully incorporated into the porous TiO_2_. Therefore, the CDs coated in the porous TiO_2_ are successfully fabricated. The TEM image also confirms the network mesoporous architecture of porous interconnected channels with incorporated carbon dots (Figure 4a). The HRTEM image of the TiO_2_/CDs shows a clear lattice spacing of 0.35 nm, which corresponds to the (101) plane of the anatase TiO_2_ (Figure 4b) and the interface of amorphous carbon dots and continuous lattice fringes of the TiO_2_ (Figure 4b).

The surface area and the pore characteristics of the TiO_2_ product were investigated by measuring the nitrogen isotherms (adsorption–desorption loop), as shown in Figure 5a. The hysteresis loop of the prepared TiO_2_ demonstrated a typical type IV isotherm, revealing the characteristics of the mesoporous materials. According to a previous report [42], the IV isotherm was normally observed in mesoporous materials with pores wider than 4 nm. Herein, the hysteresis ring is observed in Figure 5a, which may have arisen from capillary evaporation and condensation occurring under different pressures. The specific surface area of the porous TiO_2_ was calculated to be 235.86 m^2^/g, which is much larger than the commercial TiO_2_ (P25) (50 m^2^/g, data provided by the manufacturer). The pore size distribution of the sample was determined by the Barret–Joyner–Halenda (BJH) equation [43], which suggested a well-defined porous structure (Figure 5b). The pore size distribution shows that the majority of the pores are around 5 nm in diameter, consistent with our HRTEM results. The BET results also show that the mesoporous channels remain open. Such open mesoporous architecture, with the connected pore system and large surface area, plays an important role in catalyst design for the ability to improve the molecular transport of reactants and products [27]. After integration with the CDs, the surface area decreased to 124.923 m^2^/g. The average pore size reduced to 1.07 nm, revealing that the CDs were incorporated into the large pores of the TiO_2_.

The UV-Vis absorption spectra of the pure CDs, the pure porous TiO_2_ and the TiO_2_/CDs are shown in Figure 5c,d. Pure CDs exhibited two strong absorption peaks in the UV region, tailing into the visible range until λ = 550 nm. The curves of the porous TiO_2_ and TiO_2_/CDs showed strong light absorption at the UV and visible wavelength of 200–800 nm. Notably, compared with the pure porous TiO_2_, the TiO_2_/CDs showed a wider peak and higher absorption intensity, indicating a significant enhancement of light absorption. The peak position of the porous TiO_2_ was located at ~290 nm, while that of the TiO_2_/CDs shifted to ~365 nm. Similarly, the P25 TiO_2_ with the CDs incorporation demonstrated a broader peak, a higher absorption intensity and a longer wavelength of the peak position than the pure P25 TiO_2_, as can be seen in Appendix A. For both our synthesized TiO_2_ and the commercial P25 TiO_2_, their light absorbance was extended and enhanced with the import of CDs. The reason for the enhancement of the UV and the visible light absorption can be attributed to two parts: (1) the TiO_2_/CDs performed like a “dyade” structure, where the TiO_2_ and the CDs may form a joint electronic system, which gives rise to synergistic properties [6]; (2) the CDs themselves, which can absorb UV light and parts of visible light [10]. The above results indicate that we demonstrated the design of photocatalysts to enable us to harness the full spectrum of sunlight. The porous TiO_2_, with its high surface area, possessed better absorbance of the UV light and even the visible light than the commercial P25. Moreover, the sensitization of the TiO_2_ with the CDs could significantly enhance their UV-Vis light absorption and reduce the extinction in the IR region. The as-prepared pure CDs exhibited an excitation-dependent down-conversion photoluminescent (PL) behavior (Appendix A). It is interesting that no position shift of PL emission peaks of the TiO_2_/CDs upon excitation from 280 nm to 800 nm occurs (Figure 5d). The peak is around 512 nm, which further proves the TiO_2_/CDs photocatalyst can absorb a wide spectrum of sunlight. Moreover, the maximum emission for the TiO_2_/CDs was found at 460 nm excitation, which further confirmed that the CDs attached on the porous TiO_2_ have a preferential absorbance at 460 nm. This can also amplify the photocatalytic property of the composite.

The photocatalytic activities of the prepared porous TiO_2_ and the porous TiO_2_/CDs samples were evaluated by the degradation reaction of the organic dye Rh-B. The concentration of Rh-B was calculated as a function of the irradiation time by measuring the absorbance intensity changes at 554 nm with a UV-Vis spectrophotometer. To eliminate the influence caused by adherence of the Rh-B on the surface of the composite NPs, the samples were permitted to be fully saturated with the Rh-B solution in a dark environment under magnetic stirring until the Rh-B concentration of the supernatants remained constant. After full adsorption in the pores, the photocatalytic activities of the samples were evaluated by measuring the time-dependent photodecomposition of the Rh-B aqueous solutions upon irradiation, as shown in Figure 6a,b. The absorption intensity at λ = 553 nm decreased significantly for both sample systems when prolonging the irradiation time. It was found thatTiO_2_ incorporated with CDs worked more effectively for Rh-B degradation. After 90 min of irradiation, Figure 6c shows that around 40% of Rh-B was still maintained in the pure TiO_2_ system; however, only around 10% of Rh-B was not degraded in the TiO_2_/CD system. The degradation rates of the Rh-B dye over TiO_2_ with and without CD incorporation are shown. The corresponding degradation rates of the Rh-B dye over different samples are displayed in Figure 6d, indicating a higher degradation rate of the TiO_2_/CDs than that of the TiO_2_. In addition, contrast experiments were carried out using commercial P25 as photocatalysts, and the same procedure for the reduction of Rh-B was employed (Figure 7). As illustrated in Figure 7a,b, the absorption intensity decreased significantly for both sample systems when prolonging the irradiation time. The P25 TiO_2_ with CD incorporation demonstrated a better degradation efficiency and a higher degradation rate (Figure 7c,d). The results indicate that around 22% of the Rh-B was in the P25/CDs sample, while around 50% in the P25 samples was left after 90 min of irradiation (Figure 7c).

The above analysis reveals that both the synthesized TiO_2_ and the commercial P25 TiO_2_ samples with CDs showed higher photocatalytic activity of the corresponding bare TiO_2_ materials upon the light irradiation, thus proving that indeed a special behavior is to be expected due to the intrinsic properties of the CDs and the possible interaction between the CDs and the TiO_2_. The possible mechanism could be explained as follows. Firstly, the CD incorporation could increase the amount of total light absorption, which should be from both CD and Ti species. As demonstrated in Figure 5c, the absorption edge of the TiO_2_ exhibits red-shifted absorption after bonding with the CDs, indicating that the hybrid TiO_2_/CDs particles have an extended light absorption spectrum and can absorb more light during the photocatalytic process. Secondly, the CDs can create an intragap in the localized states of the C 2p situated around the valence band edge of the TiO_2_ (Figure 8) [30], and the NIR light irradiated on the CDs can be up-converted to visible light (Figure 5d). Therefore, the Ti species can also be excited by the visible light and even the NIR light, so as to produce electron–hole pairs and generate ·O_2_^−^ and ·OH. Thirdly, the CDs are able to serve as electron acceptors and donors [20], wherein the photo-induced electrons can transfer from the CDs to the TiO_2_ surfaces, and then the redundant electrons on TiO_2_ can transfer back to the CDs, as illustrated in Figure 8. To be more precise, on the one hand, the CDs can serve as electron reservoirs to trap the photo-induced electrons from the TiO_2_, facilitating the efficient separation of electrons and holes and thereby improving their photocatalytic activity. On the other hand, the CDs could work as photosensitizers to TiO_2_, sensitizing the TiO_2_ through the possible formed Ti–O–C bond between the TiO_2_ and CDs, and inject the photo-generated electrons to the conduction band of TiO_2_ [44]. When the electron is transferred to the surface-absorbed O_2_, active species of ·O_2_^−^ will form. Consequently, the integration of CDs and TiO_2_ could efficiently promote the transfer of photo-generated electrons, reducing the electron–hole recombination rate, and then greatly enhancing the generation of active species on the surface [30]. Furthermore, the synthesized TiO_2_/CDs demonstrated better photocatalytic activity than the P25/CD sample, demonstrating that the microstructure of particles plays an important role in their efficient photocatalytic activity. Specifically, mesopores in the TiO_2_ particles influenced the photocatalytic effect to a great extent, which was mainly due to two aspects: (1) a large surface area resulting from the mesoporous structure, providing plenty of active sites and enabling a better adsorption of the Rh-B molecules; (2) mesoporous channels providing paths for dye molecules to diffuse easily and reach the active sites [45]. The larger surface area of the porous TiO_2_ was proved by the BET data (Figure 5a,b). All of the above proposed causes contributed to the final excellent photocatalytic activity of the porous TiO_2_/CDs hybrid materials.

## 4. Conclusions

The photo-degradation of Rh-B demonstrated an excellent photocatalytic activity of the TiO_2_/CD hybrid particles, which was much higher than the pristine porous TiO_2_. With CD incorporation, P25 also showed an increased photocatalytic activity than that without the CDs. It is believed that the new chemical and optical properties of the TiO_2_/CDs nanohybrid were introduced by CDs, which synchronously made a great contribution to the enhanced photocatalytic activity. The photocatalytic differences between the mesoporous TiO_2_/CDs and the P25/CDs indicate that the nanostructure of the TiO_2_ particles played a valuable role in their photocatalytic activities. Compared with the commercial P25, our synthesized porous TiO_2_, with its large amounts of pores and extremely high surface areas (235.86 m^2^/g), displayed a better behavior in the photo-degradation of Rh-B.

## Data Availability

Not applicable.

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
