# Peer review of "Porous TiO2/Carbon Dot Nanoflowers with Enhanced Surface Areas for Improving Photocatalytic Activity"

_nanomaterials, 2022, doi:10.3390/nano12152536_

Round 1

Reviewer 1 Report

In the article “Porous TiO2/Carbon dots Nanoflowers with Enhanced Surface Areas for Improving Photocatalytic Activity”, a photocatalytic behaviour of as-synthesized TiO2 and TiO2/CDs samples towards the degradation of rhodamine B (Rh-B) was studied. The Authors tried to reveal the influence of Carbon dots doping on the microstructure and photocatalytic properties of the TiO2/CDs hybrid particles. However, a few comments should be providing.

 1.      In the title of the article the term nanoflowers is used, which shows the specific structure of the hybrid particles. However, there is no confirmation of this term in the article. The main supporting information (SEM-images and EDX spectra) is included in the Supplementary Materials at www.mdpi.com/xxx/s1, which may not be available. I believe that these materials should be presented in the article.

2.      Authors should explain in more detail what it means “self-assembly method” of the TiO2/CDs particles formation.

3.      From the text of the article, it is not clear enough how the structure of “nanoflowers” affects the photocatalytic properties of TiO2/CDs hybrid particles.

4.      In line 164, the Authors wrote, "anatase has a higher surface density of surface hydroxyls." Explain exactly what they mean.

5.      In the Figure 5 hydroxyl ion is shown without negative charge.

Author Response

In the article “Porous TiO2/Carbon dots Nanoflowers with Enhanced Surface Areas for Improving Photocatalytic Activity”, a photocatalytic behaviour of as-synthesized TiO2 and TiO2/CDs samples towards the degradation of rhodamine B (Rh-B) was studied. The Authors tried to reveal the influence of Carbon dots doping on the microstructure and photocatalytic properties of the TiO2/CDs hybrid particles. However, a few comments should be providing.

  1. In the title of the article the term nanoflowers is used, which shows the specific structure of the hybrid particles. However, there is no confirmation of this term in the article. The main supporting information (SEM-images and EDX spectra) is included in the Supplementary Materials at www.mdpi.com/xxx/s1, which may not be available. I believe that these materials should be presented in the article.

Answer: Thanks for your suggestions. The SEM-images and EDX spectra were moved to main text.

  1. Authors should explain in more detail what it means “self-assembly method” of the TiO2/CDs particles formation.

Answer: “self-assembly method”is simply physical mixing of TiO2 and CDs. The detailed method was as following: “The as-prepared mesoporous TiO2 particles were coated with CDs by soaking them in the CD solution in ethanol for 24 h before removing and rinsing with clean ethanol to remove any poorly adhered CDs.”

  1. From the text of the article, it is not clear enough how the structure of “nanoflowers” affects the photocatalytic properties of TiO2/CDs hybrid particles.

Answer: Nanoflowers structure can enlarge porprosity of TiO2/CDs hybrid particles. Such TiO2/CDs hybrid particles with large porprosity are normally possess high surface areas which tends to increase the effective contact areas between the pollutants and the active surfaces of photocatalytic particles.

  1. In line 164, the Authors wrote, "anatase has a higher surface density of surface hydroxyls." Explain exactly what they mean.

Answer: According to the previous report (Adv. Mater. 2009, 21, 2233-2239.), they think anatase has a higher area density of surface hydroxy. This high area density of surface hydroxy favors to capture CDs and slows the recombination of photogenerated electron–hole pairs.

  1. In the Figure 5 hydroxyl ion is shown without negative charge.

Answer: In fact, hydroxyl radical was proposed in the Figure 8.

Reviewer 2 Report

In this paper, the authors synthesized and tested a carbon dots modified TiO2 nanoparticles with flower-like mesoporous structure.

The paper is properly divided in sections and sub-sections but needs some corrections before its publication in the journal.

-        Did the authors calculate the dimensions of the crystallite of the prepared TiO2?

-        The authors should introduce all the acronyms at the first appearance in the text;

-        Why did the authors perform their experimental tests for 90 minutes? If the tests have a longer duration, did the data relevant to porous TiO2/CDs (figure 4) reach a higher degradation? The data in figure 4c seems to indicate that not a stationary condition was reached;

-        In my opinion, the authors could add the comparison with the P25 based catalysts in the paper: this could be interesting for the reader;

-         

Author Response

In this paper, the authors synthesized and tested carbon dots modified TiO2 nanoparticles with flower-like mesoporous structure. The paper is properly divided in sections and sub-sections but needs some corrections before its publication in the journal.

  1. Did the authors calculate the dimensions of the crystallite of the prepared TiO2?

 Answer: Generally speaking, the dimensions of the crystallite of the prepared TiO2 can be calculated from XRD data. According to Scherrer formula in the form “Dhkl = 0.89λ/(βcosθ) ”, where Dhkl represents diameter of crystallite. However, the dimensions of the crystallite is not focus of our attention. Hence, we have not calculated the dimensions of the crystallite of the prepared TiO2.

  1. The authors should introduce all the acronyms at the first appearance in the text;

Answer: We have introduced all the acronyms with yellow mark at the first appearance in the revised text

  1. Why did the authors perform their experimental tests for 90 minutes? If the tests have a longer duration, did the data relevant to porous TiO2/CDs (Figure 4) reach a higher degradation? The data in figure 4c seems to indicate that not a stationary condition was reached;

Answer: Thanks for your suggestions. If the tests have a longer duration, the degradation will be balance. Hence 90 min was selected in this system.

  1. In my opinion, the authors could add the comparison with the P25 based catalysts in the paper: this could be interesting for the reader;

Answer: Thanks for your suggestions. We add the performance of P25 based catalysts in main text. As illustrated in Figure 7a,b, The absorption intensity decreased significantly for both sample systems when prolonging the irradiation time. P25 with CDs incorporation demonstrated a better degradation efficiency and a higher degradation rate (Figure 7c,d). The results indicated that around 22% of Rh-B in P25/CDs sample while around 50% in P25 samples were left after 90 mins irradiation (Figure 7c).

Figure 7. UV-Vis spectra of RhB-degradation with (a) P25 and (b) P25/CDs under UV light at different irradiation time. (c) Photocatalytic degradation curves of P25 and P25/CDs at different time under UV light irradiation. (d) Pseudo-first-order fitted degradation of RhB by P25 and P25/CDs.

Round 2

Reviewer 2 Report

The authors improved the manuscript, but minor issues are still present:

1. Figure 5a is not addressed at line 212;

2. May the authors add a comment regarding the hysteresis in figure 5a?

Author Response

Reviewer 2

The authors improved the manuscript, but minor issues are still present:

  1. Figure 5a is not addressed at line 212.

Answer: Thanks for your suggestions. I revised it as “Figure 5a” in new manuscript.

  1. May the authors add a comment regarding the hysteresis in figure 5a?

Answer: According to previous report (Chem. Soc. Rev., 2017,46, 389-414), the IV isotherm was normally observed in mesoporous materials with pores wider than 4 nm. Herein, hysteresis ring is observed in Figure 5a, which may be arisen from capillary evaporation and condensation occurring under different pressures.
